# Gut Microbiota Profile of Obese Diabetic Women Submitted to Roux-en-Y Gastric Bypass and Its Association with Food Intake and Postoperative Diabetes Remission

**DOI:** 10.3390/nu12020278

**Published:** 2020-01-21

**Authors:** Karina Al Assal, Edi Prifti, Eugeni Belda, Priscila Sala, Karine Clément, Maria-Carlota Dao, Joel Doré, Florence Levenez, Carla R. Taddei, Danielle Cristina Fonseca, Ilanna Marques Rocha, Bianca Depieri Balmant, Andrew Maltez Thomas, Marco A. Santo, Emmanuel Dias-Neto, João Carlos Setubal, Jean-Daniel Zucker, Giliane Belarmino, Raquel Susana Torrinhas, Dan L. Waitzberg

**Affiliations:** 1Department of Gastroenterology, FMUSP, School of Medicine, University of São Paulo, Digestive Surgery Discipline (LIM 35), São Paulo 01246-903, Brazil; sala.priscila@gmail.com (P.S.); daniellecf_25@hotmail.com (D.C.F.); ilanna.marques@gmail.com (I.M.R.); biancadepieribalmant@hotmail.com (B.D.B.); santomarco@uol.com.br (M.A.S.); gilianebelarmino@gmail.com (G.B.); rtorrinhas@gmail.com (R.S.T.); dan@ganep.com.br (D.L.W.); 2Sorbonne Universités, IRD, Unité de Modélisation Mathématique et Informatique des Systèmes Complexes (UMMISCO), F-93143 Bondy, France; edi.prifti@ird.fr (E.P.); jean-daniel.zucker@ird.fr (J.-D.Z.); 3Institute of Cardiometabolism and Nutrition (ICAN) Integromics Team, Pitié-Salpêtrière Hospital, 75013 Paris, France; e.belda@ican-institute.org (E.B.); carlota.dao@tufts.edu (M.-C.D.); 4Assistance Publique Hôpitaux de Paris, Nutrition Department, Pitié-Salpêtrière Hospital, 75013 Paris, France; karine.clement@psl.aphp.fr; 5Sorbonne Université, INSERM, NutriOmics Research Unit, 75013 Paris, France; 6Université Paris-Saclay, INRA, MetaGenoPolis, AgroParisTech, MICALIS, 78350 Jouy-en-Josas, France; joel.dore@inra.fr (J.D.); florence.levenez@inra.fr (F.L.); 7Department of Clinical Analysis and Toxicology, School of Pharmaceutical Sciences, University of São Paulo, São Paulo 05508-000, Brazil; crtaddei@usp.br; 8Medical Genomics Laboratory, CIPE/A.C. Camargo Cancer Center, São Paulo 01525-001, Brazil; andrew.thomas@accamargo.org.br (A.M.T.); emmanuel@accamargo.org.br (E.D.-N.); 9Laboratory of Neurosciences (LIM-27) Alzira Denise Hertzog Silva, Institute of Psychiatry, Faculdade de Medicina, Universidade de São Paulo (USP), São Paulo 05403-010, Brazil; 10Department of Biochemistry, Institute of Chemistry, University of São Paulo, São Paulo 05508-220, Brazil; setubal@iq.usp.br; 11Biocomplexity Institute of Virginia Tech, Blacksburg, VA 24061, USA

**Keywords:** bariatric surgery, type 2 diabetes mellitus, gut microbiota

## Abstract

Gut microbiota composition is influenced by environmental factors and has been shown to impact body metabolism. Objective: To assess the gut microbiota profile before and after Roux-en-Y gastric bypass (RYGB) and the correlation with food intake and postoperative type 2 diabetes remission (T2Dr). Design: Gut microbiota profile from obese diabetic women was evaluated before (*n* = 25) and 3 (*n* = 20) and 12 months (*n* = 14) after RYGB, using MiSeq Illumina-based V4 bacterial *16S rRNA* gene profiling. Data on food intake (7-day record) and T2Dr (American Diabetes Association (ADA) criteria) were recorded. Results: Preoperatively, the abundance of five bacteria genera differed between patients with (57%) and without T2Dr (*p* < 0.050). Preoperative gut bacteria genus signature was able to predict the T2Dr status with 0.94 accuracy ROC curve (receiver operating characteristic curve). Postoperatively (vs. preoperative), the relative abundance of some gut bacteria genera changed, the gut microbial richness increased, and the *Firmicutes* to *Bacteroidetes* ratio (rFB) decreased (*p* < 0.05) regardless of T2Dr. Richness levels was correlated with dietary profile pre and postoperatively, mainly displaying positive and inverse correlations with fiber and lipid intakes, respectively (*p* < 0.05). Conclusions: Gut microbiota profile was influenced by RYGB and correlated with diet and T2Dr preoperatively, suggesting the possibility to assess its composition to predict postoperative T2Dr.

## 1. Introduction

Obesity is a worldwide epidemic associated with an increasing prevalence of type 2 diabetes mellitus (T2D) and its associated comorbidities. Over the past 10 years, animal and human studies have shown that obesity and its comorbidities are accompanied by changes in intestinal microbiota composition [1,2]. Long-term gut dysbiosis could promote epigenetic and metagenomic changes, which will further contribute to diabetes development [3]. Several factors associated with development of obesity, such as diet and lifestyle, can influence the gut bacteria profile, which in turn may contribute to obesity development [4,5]. For instance, high-fat diets can alter gut microbiota composition [6] and this change may be associated with obesity [5].

Whether altered gut microbiota composition has a subordinate or effector relationship with obesity remains to be better understood. However, evidence has suggested that commensal enteric bacteria may modulate the endogenous metabolism by influencing host energy harvest [7] and the release of intestinal hormones that regulate peripheral metabolism, such as insulin sensitivity, glucose tolerance, fat storage, and appetite [8]. Furthermore, gut microbiota may be a key factor influencing the complex regulation of body weight [1,2]. In germfree mice the fecal transplant of obese animals induced weight gain [1,9]. In humans, reduced microbial gene richness and diversity were associated with increased BMI (body mass index), low grade-inflammation, and insulin resistance [10,11].

Bariatric surgery is an integral part of severe obesity treatment and, particularly, the Roux-en-Y gastric bypass (RYGB) surgery is effective in inducing fast and significant weight loss with improvement or remission of obesity related complications in most of patients [12,13]. RYGB is quite efficient for T2D resolution and considered by the Metabolic Surgery Guideline as the elective surgery of choice for improving metabolic control in diabetic patients [14,15,16]. After RYGB, experimental and clinical studies have reported changes in gut microbiota composition and diversity including a reduction in the Firmicutes/Bacteroidetes ratio (rFB) [2,17]. Furthermore, an increase in *Proteobacteria* species has been consistently observed in humans [13,14,17,18] and rodents [19,20] following bariatric procedures.

RYGB-induced changes in microbiota are usually associated with weight loss and metabolic improvements [12,17,21]. When evaluating the fecal microbiota profile of obese diabetic patients before and after RYGB, Murphy R et al. [22] found increased preoperative levels of *Desulfovibrio* in patients without postoperative T2D remission when compared to patients who achieved this metabolic improvement. This finding, if confirmed, could impact the predictive outcome of RYGB for T2D resolution.

The gastric volume reduction, included in the RYGB technique, decreases dramatically the food intake amount. Individual alterations in diet can modify the gut microbiota [23] and should be taken into account when considering gut microbiota changes after bariatric procedures. Our study was designed to assess the gut bacteria profile from obese diabetic patients before and after RYGB and its correlation with food intake. Furthermore, we tested possible correlations between gut microbiota profile and postoperative T2D remission, to assess whether gut bacteria may influence this metabolic outcome.

## 2. Materials and Methods

### 2.1. Ethical Issues

This study was performed in accordance with the ethical standards of the World Medical Association (WMA) Declaration of Helsinki. The study protocol was approved by the local ethics committee (Reference: CAPPesq 1011/09) and forms part of a major clinical trial (www.clinicaltrials.gov NCT01251016), which patient recruitment procedures and the partial study design were previously published elsewhere [24]. Written informed consent was obtained from each patient prior to recruitment.

### 2.2. Patients

Obese women (*n* = 25) who were candidates for RYGB were recruited from the Coloproctology and Digestive Surgery Service of Hospital das Clínicas of the University of São Paulo, School of Medicine (HC-FMUSP). The inclusion criteria were as follows: (i) aged 18–60 years; (ii) BMI ≥ 35 kg/m^2^; (iii) confirmed diagnosis of T2D with a fasting glucose ≥ 126 mg/dL; (iv) glycated hemoglobin (HbA1c) ≥ 7%, and (v) current use of oral anti-diabetic drugs. Patients were excluded for the following reasons: (i) *Helicobacter pylori* infection; (ii) diagnosis of thyroid or hepatic disease; (iii) use of antibiotics during the month preceding fecal sample collection; (iv) use of probiotics or prebiotics, and (v) current or recent participation in another interventional study. All patients underwent open RYGB surgery without silicon rings with standardized lengths for biliary-pancreatic loops (50–60 cm) and feed handles (100–120 cm). All the analyses and data collection of this study were performed before and 3 and 12 months after RYGB.

### 2.3. Metabolic Markers

Blood samples were collected in tubes containing ethylenediaminetetraacetic acid (EDTA) after a 12-h fast. Plasma was obtained by centrifugation (2800 rpm at 4 °C for 10 min) and was used to determine the levels of C-peptide, glucose, insulin, HbA1c, high-density lipoprotein (HDL), low-density lipoprotein (LDL), very-low-density lipoprotein (VLDL), and triglycerides. These analyses were performed at the Clinical Laboratory of the HC-FMUSP using enzymatic methods (glucose and lipid profiles), liquid chromatography (HbA1c), and electrochemiluminescence (insulin and C-peptide). Data were used to calculate homeostatic model assessment of insulin resistance (HOMA-IR) levels and to identify hypercholesterolemia, hypertriglyceridemia, and metabolic syndrome, according to the criteria of the International Diabetes Federation (IDF) [25]. The intake of medication for metabolic control (i.e., diabetes, hypertriglyceridemia, hypertension) was also recorded.

### 2.4. T2D Remission

Total T2D remission was defined in accordance with American Diabetes Association (ADA) criteria that considers the absence of oral anti-diabetic medication intake or other anti-diabetic procedures along with fasting glucose < 100 mg/dL and HbA1c < 42.1 mmol/mol for at least one year after surgery [26,27].

### 2.5. Anthropometric and Body Composition Variables

Current body weight was measured with the patient standing barefoot in the center of a calibrated electronic platform scale (Life Measurement Instruments, Concord, CA, USA) wearing only underwear. For height assessment (in cm), we used a stadiometer (Sanny^TM^, São Paulo, SP, Brazil) with the patient standing barefoot with heels together, spine erect, and arms extended next to the body. Waist (narrowest diameter between the xiphoid process and the iliac crest) and hip (widest diameter over the greater trochanters) measurements were determined as circumferences by adjusting a tape in the horizontal plane to derive the waist-to-hip ratio (WHR) [28,29].

### 2.6. Food Intake

A seven-day food record was applied one week before stool sample collection. The amount of food consumed was recorded in terms of cooking units (such as tablespoons and cups), guided by illustrations from a manual, which was offered to all the participating patients [30]. The mean calorie, carbohydrate, protein, fat, fiber, and vitamin B12 intake was estimated by using Virtual Nutri Plus^®^ software [31] and two food chemical composition tables commonly used in Brazil: Tabela Brasileira de Composição de Alimentos-TACO [32] and Tabela de Composição de Alimentos: Suporte para Decisão Nutricional (Sônia Tucunduva) [33]. The fiber/lipid intake ratio (rF/L) was also calculated by dividing the mean of seven days record of fiber intake by the mean of seven days record of fat intake for each patient at each preoperative and postoperative time points studied.

### 2.7. Microbiome Profiling

All patients collected their own fecal samples at home using a specimen collection system (Fisher Scientific, Hampton, NH, USA). Samples were frozen at −20 °C and transported in containers with dry ice to our laboratory by a courier service specialized in the transport of biological samples under controlled temperatures. Fecal samples were aliquoted immediately (without thawing) and stored at −80 °C. The frozen aliquots were then shipped to MetaGenoPoliS at Jouy-en-Josas, France (http://www.mgps.eu), where total DNA was extracted and stored at −20 °C, as detailed in the International Human Microbiome Standards (IHMS) SOP06 (http://www.microbiome-standards.org).

The V4 region of the *16S rRNA* gene of the microbiota was amplified using the two following primers [34]:

V4F: TCGTCGGCAGCCAGTGATGTGTATAAGAGACAGGTGCCAGCMGCCGCGGTAA.

V4R: GTCTCGTGGGCTCGGAGATGTGTATAAGAGACAGGGACTACHVGGGTWTCTAAT.

Amplifications were generated in two steps using a custom preparation protocol (Illumina Inc., San Diego, CA, USA). The first amplification was performed in duplicate, and the amplicons were pooled in the first clean-up. Pooled samples were loaded onto the Illumina MiSeq clamshell-style cartridge kit v2 at 500 cycles for paired-end 250 sequencing at a final concentration of 12 pM. The library was clustered to a density of approximately 800 k/mm^2^. The MiSeq platform provided image analysis, base calling, and data quality assessment (Faculdade de Ciências Farmacêuticas, Universidade de São Paulo, São Paulo, Brazil).

### 2.8. Bioinformatics and Statistical Analysis

Sequencing data were analyzed using Mothur software version 1.36 [35]. Paired-end Illumina MiSeq V4 reads were assembled and filtered according to the Mothur MiSeq SOP (http://www.mothur.org/wiki/MiSeq_SOP) as described previously [34]. In brief, contigs, or sets of overlapping DNA segments, were created by overlapping Read1 and Read2, and V4 amplification primers were identified (maximum of two mismatches) and trimmed. Contigs were then screened to remove sequences divergent from the expected range (between 250 and 275 bp), those with ambiguous base calls, and those with misalignments to the V4 region. Unique sequences and their frequency in each sample were identified and followed by a preclustering algorithm that was used to remove ‘noise’ from sequences within each sample [36]. Sequences were screened for chimeras using the UCHIME algorithm [37], and a naive Bayesian classifier was applied to each sequence using the Ribosomal Database Project (RDP) *16S rRNA* gene training set (v.9), and an 80% confidence threshold. Using Mothur, we filtered out sequences belonging to non-Eubacterial domains. Finally, sequences were split into groups according to their taxonomy at the level of order, and then operational taxonomic units (OTUs) were created at a 3% maximum dissimilarity level.

To avoid the bias associated with variable sequencing depth, each sample was downsized to 61,000 reads and then further normalized by the number of sequenced reads that were clustered in the operational taxonomic units (OTUs), yielding relative abundances that were subsequently analyzed with the MetaOMineR package [38]. The genus and OTU richness were computed using a downsizing and upsizing procedure at a final level of 100,000 reads. We computed Spearman’s correlation coefficient, and for those cases when the coefficient was 1, we performed a principal coordinate analysis (PCA) for patients providing samples at all three time points (*n* = 14). Nonparametric statistics: Mann–Whitney, Spearman correlation, and Kruskal–Wallis were applied when using microbiome data. Time-course analysis was performed using paired tests for each postoperative time point compared with baseline; this allowed us to maximize power. *p*-values were adjusted for multiple testing using the Benjamini–Hochberg (BH) procedure and were considered to be significant when the False Discovery Rate (FDR) was <5%, unless otherwise specified. Analysis and visualization were performed using the R statistical computing environment. Correlations were evaluated by using Spearman’s correlation coefficient (rho).

Predictive analyses were performed using the kernlab R package with a support vector machine (SVM) classification model (linear kernel) with default parameters and an optimized c = 10 in cross validation. The model was generated using all data points, *n* = 14 samples at baseline (six T2D nonremitted and eight T2D remitted). Due to the low number of examples, the algorithm was tested in cross validation with k = 2 (i.e., 50% training and 50% testing) drawn 50 times with different seeds (i.e., a total of 100 k-folds). Different models were created with subsets of the most correlated variables. The final model was selected with a number of features, which maximized results in cross-validation training. Graphics were built using R basic plots and the ggplot2 package.

## 3. Results

### 3.1. Patients

The baseline patient cohort was composed of 25 obese women before RYGB, with a BMI ≥ 46 kg/m^2^ (SD 5.25). As illustrated in Figure 1, some patients voluntarily withdrew from the study after 3 (*n* = 5) and 12 (*n* = 6) months of the surgery. Therefore, microbiota composition and richness, food intake, body composition and metabolic markers were assessed in 25 patients (mean age 45.80 ± 7.95) during the preoperative period and in 20 (mean age 46.80 ± 6.20) and 14 (mean age 46.50 ± 5.91) patients at 3 and 12 months after RYGB surgery, respectively. From these patients, the following data were lost: fat mass, fasting C peptide, and fasting insulin from one patient, HOMA-IR, and VLDL from two patients and diet intake record from five patients at the preoperative; HbA1c from one patient and fasting insulin, fasting C peptide and HOMA-IR from three patients at the 3-month postoperative time point; and fasting C peptide from one patient at the 6-month postoperative time point. Laboratory and microbiota profile associations with T2D remission were assessed in the 14 patients with complete data for all the periods, eight with T2D remission (mean age 48.90 ± 4.09) and six without T2D remission (mean age 43.30 ± 6.80).

### 3.2. RYGB Surgery Improved Body, Metabolic and Food Intake Profiles

Anthropometric and body composition variables were improved 3 months after surgery, including reduced body weight, reduced BMI, reduced waist circumference, and reduced body fat percentage (Table 1, Figure 2). Levels of C-peptide, glucose, insulin, HOMA-IR, HbA1c, and glucose tolerance were also improved (Table 1, Figure 2). Furthermore, 93% of our patients had stopped taking metformin (Table 1, Figure 2) and had reduced their energy intake, including reductions in carbohydrates, fats, and protein intakes (Table 1, Figure 2). At 12 months after RYGB surgery, anthropometric and body composition variables continued to improve, with a significant reduction in BMI, fat mass loss (12.8%), and lean mass gain (5%), except for one patient (Table 1, Figure 2). Medication intake for hypertension and lipid control was reduced in all patients. Energy and carbohydrate intake increased compared to the 3-month postoperative data (Table 1). No changes were observed in the fiber/lipid intake ratio (rF/L) at this time point (Table 1, Figure 2), there was a decrease in total fiber intake at both 3 and 12 months post RYGB in comparison with baseline data (Table 1, Appendix A).

We found similar results for the eight patients that achieved T2D remission and selected for having data available at the three time points studied. In these patients, most of clinical, anthropometric, body composition, and biochemical variables improved significantly 3 and/or 12 months after surgery, except for the waist/hip ratio, plasma levels of low-density lipoprotein cholesterol and total cholesterol, and the frequency of metabolic syndrome (vs. preoperative; Appendix A. The six patients without T2D remission had the same benefits, as those with T2D remission in relation to anthropometric and body composition variables, but most of clinical and biochemical variables did not change after the surgery, except by a decrease in fasting glucose, glycated hemoglobin, fasting insulin, and insulin resistance 3 months and/or 12 months after surgery and an increase of the plasma high-density lipoprotein cholesterol when comparing these postoperative time points (vs. preoperative; Appendix A). Both patients with and without T2D remission had an overall decrease in food intake 3 and 12 months after surgery. However, only patients with T2D remission did not decrease the insoluble and soluble fiber intake. Those without T2D remission did not decrease saturated fat intakes (vs. preoperative; Appendix A). When comparing the two groups of patients, significant differences were observed only for the following biochemical variables: at preoperative period fasting insulin levels was higher in patients with T2D remission than in patients without and at 12 months after surgery T2D remission patients had a decrease for hypercholesterolemia, hypertriglyceridemia, fasting glucose, and glycated hemoglobin (*p* < 0.050). No changes were observed for all the clinical, anthropometric, body composition, and food intake variables studied, although patients with T2D remission presented a borderline nonsignificant higher rF/L intake than patients without T2D remission at the 12-month postoperative time point (*p =* 0.059).

### 3.3. Patients with Total T2D Remission Displayed a Distinct Genus Signature Before RYGB Surgery

For all assessed time points, there was no significant difference in genus richness between patients with (*n* = 8) and without (*n* = 6) total T2D remission. However, at the genus level, for the preoperative time point, patients with total T2D remission had significantly lower levels of *Asaccharobacter* and *Atopobium* and higher levels of *Gemella, Coprococcus*, and *Desulfovibrio* when compared to the levels of patients without T2D remission (*p* < 0.050) (Figure 3A–E).

The univariate associations results suggested the preoperative gut microbiome to have a potential role to predict the outcome of T2D remission after surgery. We tried this hypothesis in 14 patients by building a multivariate model using Support Vector Machines (SVM). This analysis showed that with only 10 bacterial genera features, we could correctly classify all patients in terms of their T2D remission status (Figure 4B). Furthermore, when testing the generalization of the SVM approach in a cross-validation (see methods), we found that the prediction generalized fairly well, with a mean test accuracy of 0.94 in the ROC curve (receiver operating characteristic curve). (Figure 4A), a standard error of the mean <0.01, and a confidence interval ranging from 0.92 to 0.96. Since they require a larger training set due to the larger number of parameters. These 10 genera displayed differences in abundance and prevalence between the two groups (Figure 4B).

### 3.4. RYGB Surgery Was Associated with Changes in the Gut Bacteria Profile Regardless of Total T2D Remission

V4 *16S rRNA* amplicons, derived from fecal samples, generated a mean of 141,200 ± 82,194 high quality reads, with a minimum of 61,100 reads. As described above, we computed genus richness as the number of genera present in a given sample. Compared with the preoperative period, log-transformed genus richness changed following RYGB surgery, with a significant increase 3 months after surgery (*n* = 20, *p* < 0.006, Mann–Whitney paired test) and a trend towards an increase 12 months after surgery (*n* = 14, *p* < 0.060) compared with the richness at 3 months. The richness distribution at each period is shown in Figure 5A. However, the operational taxonomic unit (OTU) richness, which correlated with genus richness (*rho* = 0.69), showed similar but nonsignificant trends. The principal coordinates analyses (PCoA) (Figure 5B) showed the overall changes in the gut microbiome, as seen in the first two components; both principal components correlated with genus richness (*rho* > 0.41), with components 1 and 2 best explaining the separation between time points, which explained 36% of the genus abundance variation. Both principal components correlated with genus richness (*rho* > 0.41). This analysis showed a shift along the first axis from preoperative to 12 months after surgery and along the second axis towards 3 months after surgery, which remained stable until 12 months post-RYGB surgery. At the phylum level, the ratio of Firmicutes to Bacteroidetes (rFB) decreased significantly only 12 months after RYGB surgery when compared to the preoperative value (rFB = 1.53 vs. 3.27, *p* < 0.046 (paired Mann–Whitney)). This finding was supported by a paired test on log_10_ transformed values (Figure 5C). The microbiome at the genus level shifted considerably 3 months following RYGB surgery when compared with the preoperative baseline microbiome. At 3 months, nine different bacterial genera were more abundant: *Veillonella*, *Streptococcus*, *Gemella*, *Oribacterium*, *Atopobium*, one unclassified *Lactobacillales* genus, *Leptotrichia*, *Neisseria*, and one unclassified *Pasteurellaceae* genus. One genus (*Faecalibacterium*) was less abundant (*FDR* < 0.050; Figure 5D). At 12 months after surgery, when compared with the preoperative baseline values, we observed an increase in *Veillonella* and *Streptococcus* and a decrease in *Flavonifractor*, *Blautia*, and *Butyricicoccus* (*p* < 0.005) (Figure 5D). Only the *Oribacterium* genus significantly decreased between the 3- and 12-month time points postoperatively (*p* < 0.003). Postoperative changes in gut microbiota richness and composition were observed regardless of T2D remission status, and no specific postoperative genus signature discriminating patients who achieved this metabolic outcome were identified.

### 3.5. Genus Richness Was Associated with the Food Intake Profile

The impact of energy and nutrient (carbohydrates, protein, fiber, and lipids; rFL, and vitamin B12) intake on genus richness was analyzed at each time point of the study for each patient, in order to test individual correlations. At preoperative baseline, the genus richness had a positive correlation with the rFL (*rho* = 0.60, *p* < 0.005) (*FDR* < 0.013) (Figure 6A) and with fiber intake alone (*rho* = 0.51, *p* < 0.021) (*FDR* < 0.048) (Figure 6B). Compared to the preoperative baseline richness, the fold change in genus richness 3 months after surgery had a direct correlation with the fold change in rFL (*rho* = 0.55, *p* < 0.012, *FDR* < 0.03) (Figure 6C) and an inverse correlation with the fold change in lipids (*rho* = −0.46, *p* < 0.043, *FDR* < 0.051) (Figure 6D) as well as the fold change in vitamin B12 (*rho* = −0.47, *p* < 0.036, *FDR* < 0.051) (Figure 6E) intake. The inverse correlation between the fold change in lipid intake and the fold change in genus richness persisted until 12 months after surgery (*rho* = −0.57, *p* < 0.033) (Figure 6F). Taking all the study time points together, we searched for correlations between the intake of different dietary macronutrients and the levels of particular bacterial genera, by analyzing each patient individually. Lipid intake had a direct correlation with levels of one unclassified genus of *Acidaminococcaceae* (*rho* = 0.68, *p* < 0.001) and an inverse correlation with *Parabacteroides* levels (*rho* = −0.57, *p* < 0.008). Interestingly, protein intake showed an inverse correlation with *Akkermansia* levels (*rho* = −0.63, *p* < 0.002) and a direct correlation with levels of one unclassified *Veillonellaceae* (*rho* = 0.57, *p* < 0.008). Carbohydrate intake showed an inverse correlation with levels of *Butyricimonas* (*rho* = −0.64, *p* < 0.003) and one unclassified Proteobacteria (*rho* = −0.57, *p* < 0.009) and a positive correlation with *Rothia* levels (*rho* = 0.64, *p <* 0.003), as well as with rFL, which showed a strong positive correlation with *Collinsella* levels (*rho* = 0.72, *p* < 0.0004).

## 4. Discussion

RYGB is an efficient approach to improve severe obesity and obesity-related T2D, clinical conditions known to be accompanied by gut bacteria dysbiosis [13,14,17,18,22]. In obese diabetic patients, we here confirmed previous studies reporting significant changes in gut microbiota composition and richness following RYGB [17,22]. These changes occurred regardless of postoperative T2D remission. However, a distinct preoperative genus was observed comparing patients who achieved T2D remission and those who did not. In parallel, individual food intake, mainly in fiber and fat amount, were associated with gut microbiota richness and/or composition. Our data suggest that in T2D obese patients a particular preoperative gut microbiome may be associated, after RYGB, to glycemic homeostasis and that dietary habits may have a role in this process.

Recently, based on metabolic, clinical and genetic levels, Ahlwvist et al. classified adult-onset diabetic patients into five distinct replicable subgroups, two of which were characterized by obesity [39]. From this perspective, different phenotypes of obesity-related T2D may arise from distinct mechanisms leading to insulin resistance. One may then speculate that only patients with total T2D remission following RYGB may present significant intestinal dysbiosis as a main factor influencing glycemic homeostasis, which is reversed by the surgery. This would explain why the bacterial profile between patients with and without total T2D remission was different in the preoperative period and became more similar postoperatively. However, this hypothesis needs further exploration in extended cohorts.

In our study, the preoperative gut bacteria signature of patients with total T2D remission after RYGB included higher levels of the *Desulfovibrio* genus than those of patients without total T2D remission. Several bacteria from the *Desulfovibrio* genus are endotoxins producers and may elicit the release of inflammatory cytokines [40]. Furthermore, they have the ability to reduce sulfate to hydrogen sulfide, which can disrupt the energy metabolism of gut epithelium [41]. Taken together, these properties favor the epithelial cell death and gut barrier damage with consequent bacteria translocation [41,42]. Obesity and diabetes are characterized by a chronic low-grade inflammation that seems to involve gut bacteria dysbiosis and translocation [43]. Higher gut *Desulfobrio* concentrations are reported in pregnant woman with gestational diabetes, compared with pregnant women without gestational diabetes [44]. This observation suggests a potential role of bacteria from this genus on the diabetes development, probably by contributing to the disruption of the gut epithelial barrier.

Of interest and different from our findings, Murphy et al. [22] found a direct association between lower preoperative levels of gut *Desulfovibrio* and T2D remission after RYGB, in a previous study using a similar sample size and methodology. Fecal microbiota can vary in different regions, as observed among Chinese and European populations when studying T2D metagenomic markers [1]. Geographical and environmental aspects of the populations involved in our study and the study conducted by Murphy et al. (Brazil and New Zealand), mainly distinct food intake profile, suggest that the baseline bacterial backgrounds were different between them. For instance, red meat and fat intakes are high in New Zealand [45], probably higher than those found in patients attending a Brazilian public hospital. In fact, at the preoperative period, the patients included in Murphy et al. study presented a different food intake profile than our patients, characterized by a lower consumption of calories and general macronutrients. In our study, food intake profile influenced gut microbiota genus richness. Therefore, these different findings of *Desulfovibrio* levels may be associated with bacterial background within the genus, potentially influenced by distinct eating habits, where one contributes to glycemic homeostasis and the other contributes to the maintenance of insulin resistance.

Accordingly, we found an inverse correlation between gut bacteria richness and decreased fat intake at the postoperative period. An experimental study of mice fed a fat-rich diet reported a significant increase in *Desulfovibrio* along with a functional change in the intestinal barrier and a change in the production of microbial peptides, suggesting that *Desulfovibrio* bacteria can be sensitive to changes in fat intake [46]. It is possible that, in our study, decreased fat intake may have contributed to changes in gut bacteria background that reversed the *Desulfovibrio* prevalence observed in the preoperative gut microbiome and potentially contributed to glycemic homeostasis after RYGB. It is worth noting that decreased fat intake occurred independently of T2D remission, but even patients who did not achieve this metabolic benefit experienced an improvement of markers of glucoses homeostasis (decreased fasting glucose, glycated hemoglobin, fasting insulin, and insulin resistance) after RYGB and have a similar postoperative reduction of HOMA-IR values than patients with T2D remission (*p* > 0.050; Appendix A).

Individuals presenting low fecal microbial richness have been shown to exhibit increased adiposity, metabolic disturbances, and inflammatory phenotypes when compared to individuals with high fecal bacterial richness [10]. A direct association between fiber intake and gut bacteria richness was also observed by us and previously reported by Tap et al. [47]. In the present study, the increase in intestinal microbial richness following RYGB surgery was not associated with total T2D remission or improvements in metabolic markers. However, other studies have found a beneficial association, although these authors could not determine cause and effect [12,17]. Furthermore, in our study, only patients without T2D remission decreased total, soluble, and insoluble fiber intake and did not decrease the saturated fat intake. A lack of decrease in the fiber intake by patients with T2D remission was unexpected, facing the restrictive procedures enrolled in the RYGB technique. However, changes in food preference are reported in animals and humans after RYGB and seem to be associated with modifications in taste sensibility and, mainly, digestive motivation [48]. Our observations may have a practical effect on dietary orientation for patients after RYGB, particularly with regard to a higher fiber intake.

One feature observed by us and previously reported by others was an increase in bacteria from the Proteobacteria phylum after RYGB [22,49,50], which may be linked to changes in the pH and oxygenation of the intestinal lumen and modifications in the chewing and digestion of food, providing good conditions for the Enterobacteria growth [51]. Increased Proteobacteria is conversely associated with higher insulin resistance and fat-rich diet intake [52]. However, we and others could not find any relationship between Proteobacteria and BMI, body weight and composition, or biochemical profiles [17,50]. This conflicting evidence reinforces the impression that the potential impact of gut microbiota on clinical outcomes after RYGB may be dependent on the whole gut microbiota ecosystem response to the new surgical-induced gut environment, even at phylum level. Previous research has shown an association between increased Firmicutes/Bacteroidetes ratios and obesity [53] as well as a reduction in the Firmicutes phylum following RYGB [18,49]. Similar to Murphy et al., we observed an increase in bacteria from the Firmicutes and Actinobacteria phyla 3 months after RYGB [22]. However, although we did not find any postoperative changes between the 3- and 12-month time points in the overall abundance of Firmicutes and Bacteroidetes, we did observe a significant reduction in the Firmicutes/Bacteroidetes ratio 12 months after the surgery. This last finding has been described previously [12,18] and was associated with weight loss and metabolic improvement [2,12,17], although this association is still being debated [54].

Our study has some limitations that deserve to be highlighted. The limitations included a small number of patients, but available studies in this field have samples ranging from three to a maximum of 60 patients [12,13,17,18,22,55]. Furthermore, our obese diabetic population was composed only of women to allow for a more homogeneous sample, as fecal microbiota is known to differ by sex [56]. Although limiting our findings for males, our study has good clinical applicability, as in our public university hospital, there are far more female candidates than male candidates for bariatric surgery. Another limiting factor in our investigation was the absence of a group of obese nondiabetic patients and obese diabetic patients who did not undergo bariatric surgery and who increased their intake of dietary fiber as controls. These groups would have allowed us to identify whether the change in fecal microbiota could be due to RYGB itself or to changes in fiber ingestion. Finally, our analysis of the gut microbiota revealed only the bacterial genus taxonomy, precluding us to associate total T2D remission with changes in specific bacterial species after RYGB.

## 5. Conclusions

Supporting previous findings, RYGB increased the gut microbiota richness and altered its bacteria genus profile; however, these changes did not correlate with T2D remission. On the other hand, the gut microbiota profile was influenced by diet intake pre and postoperatively and correlated with T2D remission at the preoperative, suggesting that a particular gut microbiome profile, potentially induced by dietary habits, may be present in patients responsive to T2D remission. These findings shed some light on the possibility of evaluating the gut microbiota profile to predict T2D remission after RYGB or modulating this variable (maybe by dietary interventions) to achieve such metabolic benefit.

## Figures and Tables

**Figure 1 nutrients-12-00278-f001:**
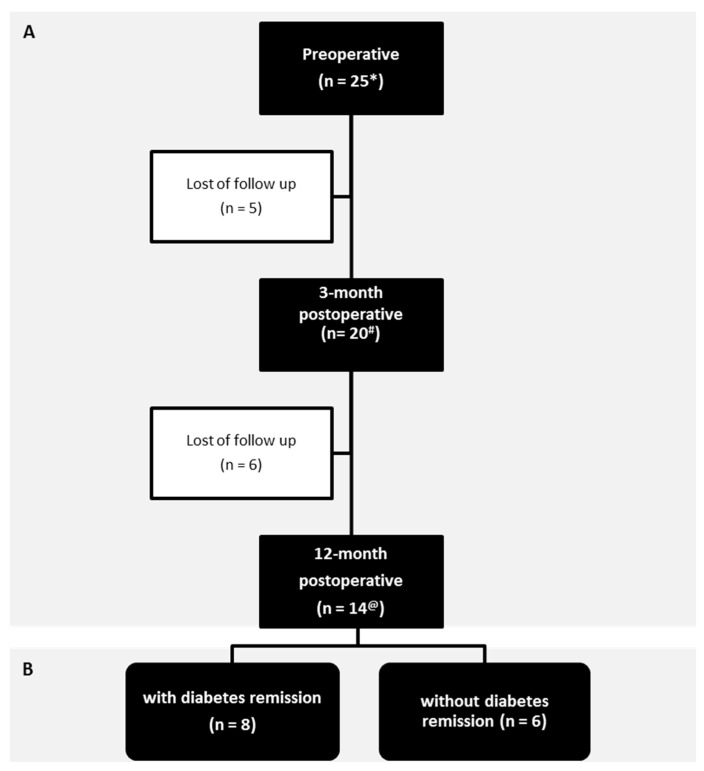
Flowchart of the patient inclusion. Stool samples were collected at each time point studied. Legend: A total of 25 patients were included in the study at preoperative, but five and six of them voluntarily withdrew 3 and 12 months after surgery, respectively. The black boxes contain the number of patients included at each time point studied with successful stool collection, in order to evaluate the microbiota composition and richness, food intake, body mass index, biochemical markers of metabolism, and the relationship of microbiota composition and richness with food intake (**A**) and the relationship of microbiota composition and richness with type 2 diabetes remission (**B**). * except for percentage of fat mass, fasting C peptide, and fasting insulin (*n* = 24), for homeostatic model assessment of insulin resistance (HOMA-IR) and very-low-density lipoprotein (VLDL) (*n* = 23) and for diet intake (*n* = 20); ^#^ except for glycated hemoglobin (HbA1c) (*n* = 19) and for fasting insulin, fasting C peptide and HOMA-IR (*n* = 17); ^@^ except for fasting C peptide (*n* = 13).

**Figure 2 nutrients-12-00278-f002:**
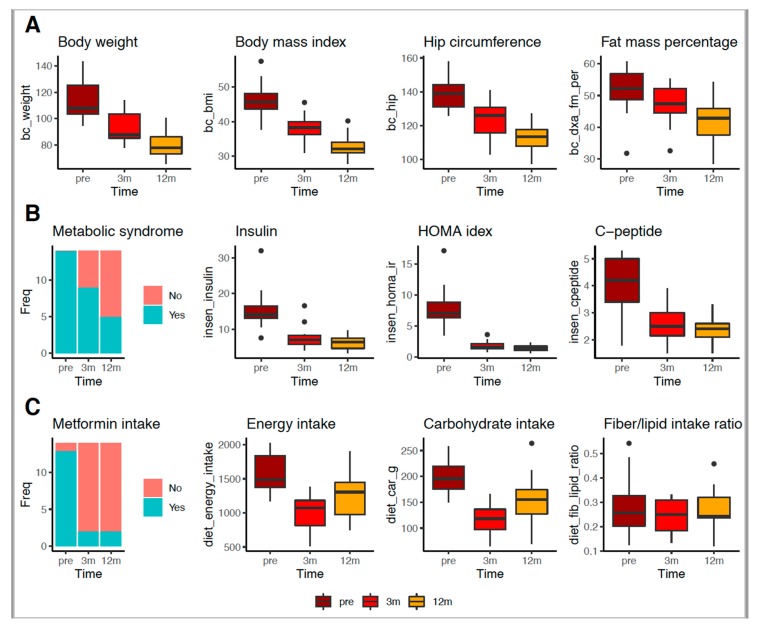
Clinical, anthropometric, body composition, and biochemical features and metformin and dietary intakes in obese patients with type 2 diabetes before and after 3 and 12 months of Roux en-Y gastric bypass. Legend: data were collected at preoperative period (brown boxes; *n* = 25), at 3 months (red boxes; *n* = 20) and 12 (orange boxes; *n* = 14) months postoperatively. Metabolic syndrome and metformin use are shown when present (salmon bars) and absent (turquoise bars) at the three observed periods. (**A**) Markers of body composition (weight, body mass index, hip circumference and fat mass), (**B**) frequency of metabolic syndrome and changes in insulin resistance markers (fasting insulin, homeostatic model assessment of insulin resistance (HOMA) index and C-peptide) were significantly lower 3 and 12 months after Roux en-Y gastric bypass than at the preoperative. (**C**) Metformin (%) and markers of food intake (energy and carbohydrates) also were significantly lower 3 and 12 months after Roux en-Y gastric bypass than at the preoperative, except for fiber/lipid ratio.

**Figure 3 nutrients-12-00278-f003:**
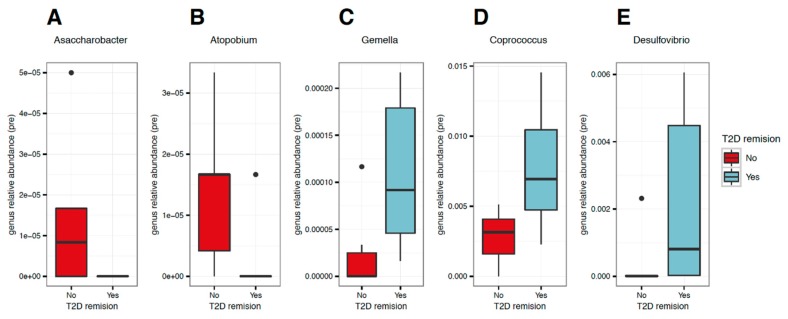
Gut bacteria genera at the preoperative period in obese patients classified according to type 2 diabetes remission after Roux-en-Y gastric bypass. Legend: comparison at the preoperative period of gut bacteria genus profile between patients classified, after Roux-en-Y gastric bypass, according to presence (blue boxes; *n* = 8) and absence of T2D remission (red boxes; *n* = 6). There was a higher relative abundance of (**A**) Asaccharobacter (*p* = 0.038) and (**B**) Stopobium (*p* = 0.047) and a lower relative abundance of (**C**) Gemella (*p* = 0.018), (**D**) Coprococcus (*p* = 0.029), and (**E**) Desulsovibrio (*p* = 0.030) in the patients with T2D remission than in patients without.

**Figure 4 nutrients-12-00278-f004:**
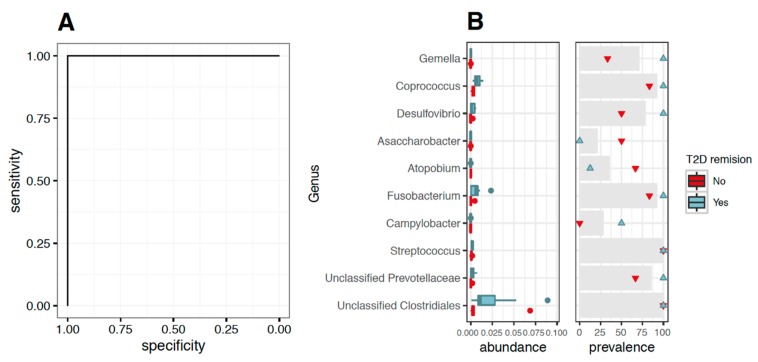
Prediction of type 2 diabetes mellitus remission after Roux-en-Y gastric bypass in obese patients based on the preoperative gut bacteria signature. Legend: (**A**) ROC plot of a classification model built with the 10 bacteria genera that most correlated with the type 2 diabetes remission status with a linear kernel showing a 100% sensitivity and specificity in predicting the outcome. (**B**). Left: boxplot graphic showing the abundance distribution of the 10 bacteria genera that most correlated with the type 2 diabetes remission status among patients with (blue boxes; *n* = 8) and without (red boxes; *n* = 6) type 2 diabetes remission. Right: prevalence of these genera in patients with (blue triangles; *n* = 8) and without (red triangles; *n* = 6) type 2 diabetes remission and in all patient samples (gray bars) is indicated as a percentage of the total follow-up cohort (*n* = 14).

**Figure 5 nutrients-12-00278-f005:**
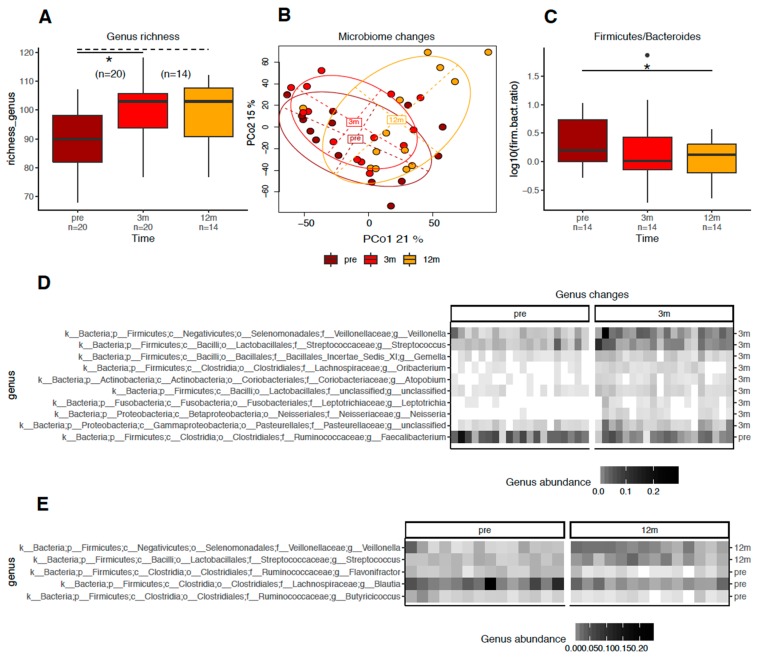
Changes of the gut microbioma composition and richness in obese patients after 3 and 12 months after Roux-en-Y gastric bypass. Legend: (**A**) boxplots showing that, in comparison to the preoperative (brown boxes), the gut bacteria genus richness increased after 3 months (red boxes) of Roux-en-Y gastric bypass (*p* = 0.006) and remained increased at the postoperative 12 months (orange boxes; *p* = 0.066 vs. postoperative 3-months). (**B**) Principal coordinates analysis (PCoA) showing an overall segregation of the gut microbiome profile (at genus level) from the preoperative (brown circle/dots) to the 3-month (red circle/dots) and 12-month (orange circle/dots) postoperative time points, according to the most discriminating factors by dissimilarity (PC01 with 21% discrimination (y axis) and PCo2 with 15% discrimination (x axis). (**C**) Boxplots showing that the Firmicutes:Bacteroidetes ratio (rFB) decreased significantly only 12 months after RYGB (*p* = 0.043). (**D**,**E**) Heatmaps of the most discriminating gut bacteria genera between the periods studied showing changes in its abundance (y axis) by patient (x axes): in comparison to the preoperative (pre; (D) and (E) left heatmaps), nine bacteria genera became more abundant and only one genus became less abundant at the 3-month postoperative time point (3 m; (D) right heatmap), while only two bacteria genera became more abundant and three bacteria genera became less abundant at the 12-month postoperative time point (12 m; (**E**) right heatmap). The intensity of square’s color (light to dark) reflects the intensity of gut bacteria genera abundance. * *p* < 0.005.

**Figure 6 nutrients-12-00278-f006:**
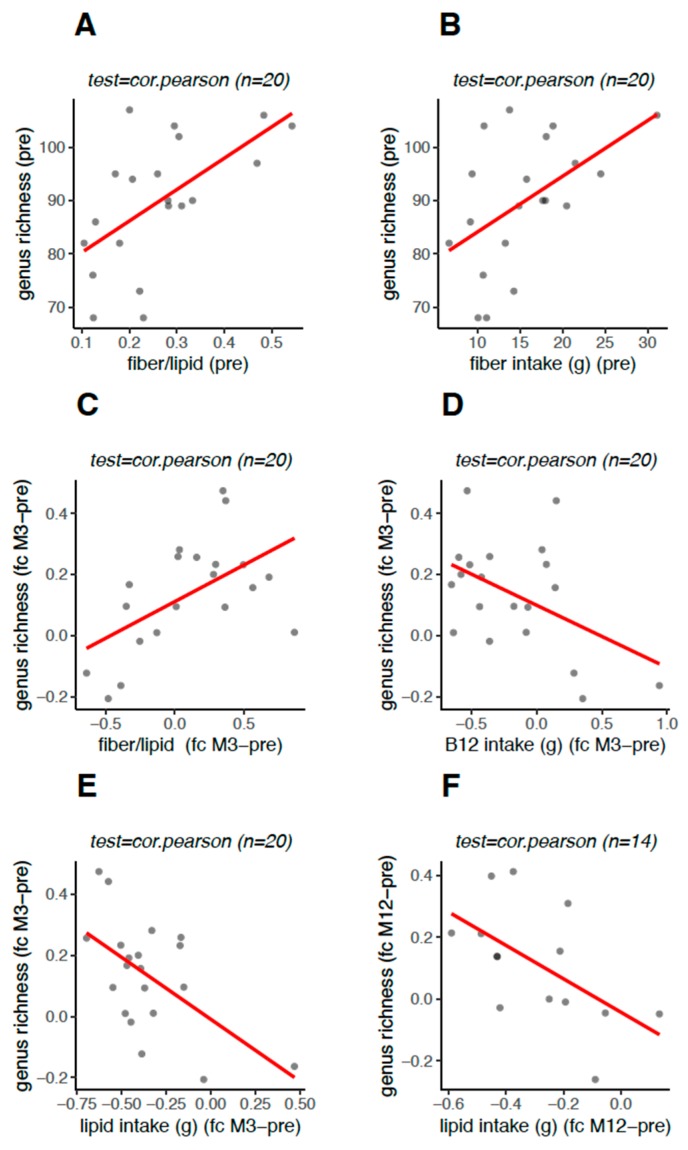
Correlation between food intake profile and gut microbiome richness in obese women with type 2 diabetes mellitus before and 3 and 12 months after Roux-en-Y gastric bypass. Legend: scatterplots indicating correlations (Pearson test) between gut microbiome richness at the bacteria genus level (x axis) and food intake profiles (y axis), where the red lines show whether these were positive (increasing lines) and negative (decreasing lines). The correlations were tested considering each patient (dots), with a direct approach for the preoperative period and between fold changes in microbiome richness and dietary intake after surgery (vs. preoperative) for the postoperative periods. Preoperativelly (pre), there was a positive correlation for fiber:lipid ratio (**A**) (*p* = 0.005, *q* = 0.013) and fiber (**B**) (*p* = 0.021, *q* = 0.048) intake with the gut microbiome richness (**A**,**B**); at the 3-month postoperative time point (M3), there was a positive correlation between the fold changes for fiber:lipid ratio (**C**) (*p* = 0.012, *q* = 0.027) and a negative correlation for vitamin B12 (**D**) (*p* = 0.036, *q* = 0.051) intake with the fold change of gut microbiome richness (**C**,**D**), respectively. At the 3-month (M3, (**E**); *p* = 0.043, *q* = 0.051) and 12-month (M12, (**F**); *p* = 0.033, *q* = 0.120) postoperative time points, there was a negative correlation between the fold change of lipid intake and the fold change of gut microbiome richness.

**Table 1 nutrients-12-00278-t001:** Clinical, anthropometric, body composition, biochemical, and daily food intake features from obese patients with type 2 diabetes, before and after 3 and 12 months of Roux en-Y gastric bypass.

Variable	Preoperative	3-Month Postoperative	12-Month Postoperative	*p* Value *	*p* Value ^#^	*p* Value ^@^
BMI (kg/m^2^)	46.40 ± 5.48	38.20 ± 4.13	32.70 ± 3.54	**<0.001**	**<0.001**	**<0.001**
Body weight (kg)	115.00 ± 15.40	94.50 ± 12.80	81.20 ± 11.10	**<0.001**	**<0.001**	**<0.050**
FM by DXA (%)	51.80 ± 6.81	46.50 ± 6.54	41.50 ± 6.72	**<0.050**	**<0.001**	**<0.050**
WC (cm)	128.00 ± 13.10	112.00 ± 12.10	103.00 ± 11.50	**<0.001**	**<0.001**	**<0.050**
HC (cm)	138.00 ± 12.10	124.00 ± 10.70	113.00 ± 7.97	**<0.001**	**<0.001**	**<0.050**
WC:HC ratio (cm)	0.93 ± 0.09	0.91 ± 0.07	0.91 ± 0.05	0.180	0.230	0.990
SBP (mmHg)	156.00 ± 30.60	139.00 ± 18.20	138.00 ± 20.70	0.063	0.079	0.940
DBP (mmHg)	102.00 ± 22.90	90.00 ± 12.50	90.90 ± 15.00	0.130	0.130	0.960
HDL (mg/dL)	44.00 ± 9.89	42.90 ± 10.20	53.40 ± 12.90	0.710	**<0.050**	**<0.050**
LDL (mg/dL)	119.00 ± 31.20	92.10 ± 29.60	85.00 ± 18.70	**<0.050**	**<0.001**	0.990
VLDL (mg/dL)	28.00 ± 8.17	21.60 ± 7.98	16.90 ± 2.96	**<0.050**	**<0.001**	**<0.050**
TC (mg/dL)	191.00 ± 33.10	157.00 ± 36.90	155.00 ± 20.00	**<0.001**	**<0.001**	0.430
Triglyc. (mg/dL)	164.00 ± 96.10	109.00 ± 40.10	83.90 ± 14.60	**<0.050**	**<0.001**	**<0.050**
Hyperchol. (%)	52.00	20.00	7.10	<0.050	**<0.050**	**0.380**
Hypertrigl. (%)	76.00	15.00	7.10	**<0.001**	**<0.001**	**0.630**
MetS (%)	100.00	70.00	36.00	**<0.05**	**<0.001**	0.080
C peptide (ng/mL)	4.05 ± 1.42	2.68 ± 0.60	2.32 ± 0.51	**<0.050**	**<0.001**	0.130
FG (mg/dL)	225.00 ± 74.00	104.00 ± 25.00	92.90 ± 16.00	**<0.001**	**<0.001**	0.300
A1c (%)	9.14 ± 1.70	6.17 ± 0.50	5.74 ± 0.54	**<0.001**	**<0.001**	0.074
FI (mUI/L)	21.80 ± 15.90	8.74 ± 3.96	6.16 ± 1.95	**<0.001**	**<0.001**	0.120
HOMA-IR (%)	11.40 ± 9.77	2.19 ± 1.34	1.40 ± 0.47	**<0.001**	**<0.001**	0.085
Metformin use (%)	84.00	10.00	14.00	**<0.001**	**<0.001**	**1.000**
Energy (kcal/day)	1700.00 ± 462.00	957.00 ± 226.00	1280.00 ± 356.00	**<0.001**	**<0.050**	**<0.050**
Protein (g/day)	71.80 ± 18.70	48.30 ± 14.10	61.20 ± 17.40	**<0.001**	0.170	**<0.050**
Carbohydrate (g/day)	213.00 ± 51.50	107.00 ± 30.80	157.00 ± 47.8	**<0.001**	**<0.050**	**<0.001**
Fat (g/day)	64.50 ± 22.20	38.60 ± 10.3	39.80 ± 8.46	**<0.001**	**<0.001**	0.780
Saturated fat (g/day)	18.50 ± 7.59	11.30 ± 3.94	11.60 ± 1.63	**<0.050**	**<0.001**	0.960
MUF (g/day)	17.60 ± 6.68	9.80 ± 3.41	10.10 ± 2.98	**<0.001**	**<0.050**	0.560
PUF (g/day)	12.40 ± 3.30	8.27 ± 2.17	6.22 ± 1.10	**<0.001**	**<0.001**	**<0.050**
Fiber (g/day)	15.50 ± 5.95	9.61 ± 3.84	10.70 ± 3.05	**<0.050**	**<0.050**	0.290
IF (g/day)	3.69 ± 1.37	2.47 ± 1.13	2.83 ± 1.09	**<0.050**	<0.064	0.310
SF (g/day)	2.43 ± 1.43	1.37 ± 0.73	1.69 ± 0.79	**<0.001**	**<0.059**	0.180
F:L ratio (g/day)	0.26 ± 0.12	0.25 ± 0.09	0.27 ± 0.08	0.860	0.460	0.620
B12 (mcg/day)	0.89 ± 0.21	0.71 ± 0.32	0.82 ± 0.46	0.053	0.064	0.850

Legend: Data expressed in mean ± standard deviation or absolute percentage. Significant differences between periods are highlighted in bold, according to the following comparisons: * 3-month postoperative (*n* = 20) versus preoperative (*n* = 25); ^#^ 12-month postoperative (*n* = 14) versus preoperative; ^@^ 3-month postoperative versus 12-month postoperative. Most variables were significantly lower 3 and/or 12 months after surgery than at preoperative, except waist:hip ratio, diastolic and systolic blood pressures, and fiber:lipid ratio and vitamin B12 intakes. Missing data: one patient for percentage of fat mass, fasting C peptide, and fasting insulin, two patients for insulin resistance and very-low-density lipoprotein cholesterol, and five patients for diet intake at the preoperative; one patient for glycated hemoglobin and for fasting insulin, and three patients for fasting C peptide and insulin resistance at the 3-month postoperative time point; one patient for fasting C peptide at the 12-month postoperative time point. A1C, glycated hemoglobin; B12, vitamin B12; BMI, body mass index; DBP, diastolic blood pressure; DXA, dual-energy X-ray absorptiometry; F:L ratio, fiber to lipid ratio; FG, fasting glucose; FI, fasting insulin; FM, fat mass; HC, hip circumference; HDL, high-density lipoprotein cholesterol; Hiperchol., hypercholesterolemia; Hypertrigl., hypertriglyceridemia; HOMA-IR, insulin resistance index; IF, insoluble fiber; LDL, low-density lipoprotein cholesterol; MetS, metabolic syndrome; MSF, monounsaturated fat; PUF, polyunsaturated fat; SBP, systolic blood pressure; SF, soluble fiber; TC, total cholesterol; Triglyc., triglycerides; VLDL, very-low-density lipoprotein cholesterol; WC, waist circumference.

## Data Availability

Nucleotide sequences used for this study have been deposited in the Sequence Read Archive (SRA) under the accession number SRP113514.

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
