# Peer review of "Gut Microbiota Profile of Obese Diabetic Women Submitted to Roux-en-Y Gastric Bypass and Its Association with Food Intake and Postoperative Diabetes Remission"

_nutrients, 2020, doi:10.3390/nu12020278_

Round 1

Reviewer 1 Report

Al-Assal et al have conducted an elegant study aimed at understanding the effect of obesity on richness of the microbial population alongside correlational analysis on type 2 diabetes and diet. 

This is an interesting study that presents important findings on the potential impact of the microbial quorum on mediating metabolic effects relevant in obesity. It is well written and well thought out, however, there are some changes that should be made in order to provide clarity and clear justification. 

Introduction:

Line 78, end of sentence..."partially between humans and mice". Please insert appropriate references.  Need to describe what is known about the metabolic effects microbiota described in the introduction.  Expand on the role of microbiota in mediating metabolic/hormonal/psychological/diet changes. Address the possibility that microbiota may mediate changes or that other aforementioned changes may alter microbial profiles. 

Methods:

Please include a schematic diagram showing study design, e.g. recruitment process, n numbers and sample collection points. 

Results:

All the figure legends are far too short and need to be expanded to properly describe what is presented in the table/figure. Please provide enough detail in the figure legend for the reader to easily understand the content. 

Figures are incorrectly numbered. Please change this. 

Table 1 - Preoperative n = 14...should this not be 25 as stated in the manuscript (section 3.1)? PS 3 months n=14...shouldn't it be 20? Please check that the N numbers in the table and manuscript are correct because at present they do not match.  Line 242, section 3.2 - fibre/lipld intake ratio...how was this calculated? Unclear from methods. Important outcome measure so please clarify.  Figure 2 - legend needs substantial expansion. The panels A, B and C are not sufficient...need to include A(i), A(ii) etc with appropriate description in the legend.  Figure 2 - Y axis of graphs require proper titles and units of measurement.  Section 3.3, line 266 - this sentence is too broad and overreaching as n = 8, so such a conclusion cannot be made. Please reword the sentence so it is more speculative.  Lines 269 - 274: was this analysis conducted on n = 8 patient samples. Figure 4: blank graph with no 'curve'. Please correct.  Figure 4, panel A: very unclear as to what is being described here as the graphs are not described in figure legend. What are the bacterial names on top of each graph referring to? Figure 3. Need to explain F/L ratio and how it was obtained/calculated. Where is data showing increased fibre ingestion post RYGB (as standalone data). Why is there difference in data points...e.g. graph F clearly has fewer N numbers. X axis titles for E and F graphs missing. Why was data separated into BMI </> 40?

Discussion:

Please state the significance of Desulfovibrio in the context of obesity and T2D. 

Reviewer 2 Report

Dear Authors, Dear Editor,

I have received the manuscript entitled "Gut microbiota profile and diabetes remission after Roux-en-Y gastric bypass in obese diabetic women" written by Karina Al Assal and colleagues in order to evaluate its value to be published in the Nutrients journal. The article describes the potential effects between metabolic improvements after Roux-en-Y gastric bypass and modulation of gut microbiota in obese diabetic women.

Generally, it is interesting quite a well written study, however, I have some doubts about the study design, presented results or conclusions. All are presented below.

First of all, I thought that this manuscript does not really match to aims and scopes of the journal. Title and abstract suggest more clinical than “nutrient” work. However, the article matches with Nutrients, just title suggests other findings then are presented in the article. In the first impression, the reader is expecting clinical paper but probably rather diet than surgery influence is presented there. For me, the discussion part does not really contain potential links with RYGB and used technique (maybe only lines 421-425), diabetes remission, etc. Thus, it could be modified. In my opinion, also the clinical side of the study is not prepared the best (some comments below).

In some moments, I am lost about the idea of the study and its aim. If we are focusing on T2D remission why there are comparisons in three time points without dividing into these groups? That kind of comparison could be an interesting addition for tests in remission/non-remission groups. An example can be Table 1 where there is only information for the whole group in three time points. In my opinion, the presentation of the data just for all time points without showing data for particular groups (remission/non-remission) is a serious lack. Maybe the aim should be also more precise?

Additionally, I think, other parameters (like ABCD score, HOMA-IR reduction) already proposed as potential predictors of diabetes remission after bariatric surgeries should be mentioned and presented for studied groups. Maybe there are any links between those parameters and the gut microbiome signature? How did they look at the beginning? There were any statistically significant differences between the studied groups at any time points?

Were any differences in e.g. BMI, HOMA-IR or its decrease, diabetes duration (both linked with diabetes remission), HbA1c?

How about receiving diabetes treatment? Patients received just metformin? I understand that not all patients got metformin at the beginning and the proportion between them was changed during follow-up. How does it look between remission and non-remission group? Because as far as I know metformin can modulate gut microbiota composition, what in my opinion could have some influences for obtained results.

I fully understand that it is hard to have a long follow-up from patients undergoing surgery… However, groups for diabetes remission were small… was the statistical power checked?

The study was performed only on women… Can be data, in Authors' opinion, sex dependent? There are already some papers present differences in obtained results and metabolic profiles of men and women in response to bariatric surgery.

Also some minor comments:

p in  p-value should be in italic

In my version of manuscripts, figures are numbered in the wrong order (Fig,1, 4, 2, 3). The order of appearing figures in the text seems to be incorrect too.

Figure 1 – I think colors should be explained in the figure caption (even as legend), not only in the text

Line 237 – where is table 2 from that reference?

Isn’t “yes/no” missed in Figure 1B with metabolic syndrome?

Figure 4 – the legend is missed. Figures aren’t really clear to read. I think each figure panel should be explained in the caption too.

Fig 4B – Is the ROC curve on maximal values or line in the picture is just invisible?

Figure 2 – Legend should be added. Panels’ description should be in the caption.

Figure 3 – Legend should be added. Panels’ description should be in the caption.

Line 384 “opposite reported by Murphy” – Obtaining opposite results is interesting, especially when similar methodology and sample size were used. Was a difference in treatment between those studies? If yes, it should be mentioned. Differences in diet, if were any, should be mentioned too. That factors (but a few others as well) can be potential causes of differences in gut microbiota signature.

I don't feel qualified to judge the English language and style. However, in my opinion, the manuscript could be checked by a native speaker/professional proofreader.

To sum up, it is an interesting study, however, it should be reconsidered for publication after major revision.

Reviewer 3 Report

The research hypothesis is interesting and the results are presented in the intelligible form. The results sheds some new light on the microbiom changes following bariatric surgery. I have no major remarks, the smaller ones are listed below.

Page 6. In Figure 1B graph demonstrating prevalence of metabolic syndrome is unclear. It would be better if you present these data in better readable form. The same applies to the Figure 1C and graph demonstrating metformin intake.

Page 7. Why the Figure 4 follows Figure 1? Figures should be numbered in consecutive order, i.e Figure 4 should become Figure 2, current Figure 2 should be numbered Figure 3 (in order of appearance), the same applies to Figure 3, which should become Figure 4.

Page 7. In current Figure 4 (future Fig. 2) there is no ROC curve (only grid). Please correct this.

Page 7. In the current Figure 4 each panel description should begin as new paragraph (it applies to description of panel B), similarly to figures' 1 and 3 descriptions or all tables should be described without separate paragraphs for each panel (like in current Figure 3).
